# Forecasting Hospital Visits Due to Influenza Based on Emergency Department Visits for Fever: A Feasibility Study on Emergency Department-Based Syndromic Surveillance

**DOI:** 10.3390/ijerph191912954

**Published:** 2022-10-10

**Authors:** Sunghee Hong, Woo-Sik Son, Boyoung Park, Bo Youl Choi

**Affiliations:** 1Department of Preventive Medicine, Hanyang University College of Medicine, Seoul 04763, Korea; 2Department of Statistics and Data Science, Graduate School, Dongguk University, Seoul 04620, Korea; 3National Institute for Mathematical Sciences, Daejeon 34047, Korea

**Keywords:** fever, influenza, emergency department, syndromic surveillance, forecast

## Abstract

This study evaluated the use of chief complaint data from emergency departments (EDs) to detect the increment of influenza cases identified from the nationwide medical service usage and developed a forecast model to predict the number of patients with influenza using the daily number of ED visits due to fever. The National Health Insurance Service (NHIS) and the National Emergency Department Information System (NEDIS) databases from 2015 to 2019 were used. The definition of fever included having an initial body temperature ≥ 38.0 °C at an ED department or having a report of fever as a patient’s chief complaint. The moving average number of visits to the ED due to fever for the previous seven days was used. Patients in the NHIS with the International Classification of Diseases-10 codes of J09, J10, or J11 were classified as influenza cases, with a window duration of 100 days, assuming the claims were from the same season. We developed a forecast model according to an autoregressive integrated moving average (ARIMA) method using the data from 2015 to 2017 and validated it using the data from 2018 to 2019. Of the 29,142,229 ED visits from 2015 to 2019, 39.9% reported either a fever as a chief complaint or a ≥38.0 °C initial body temperature at the ED. ARIMA (1,1,1) (0,0,1)_7_ was the most appropriate model for predicting ED visits due to fever. The mean absolute percentage error (MAPE) value showed the prediction accuracy of the model. The correlation coefficient between the number of ED visits and the number of patients with influenza in the NHIS up to 14 days before the forecast, with the exceptions of the eighth, ninth, and twelfth days, was higher than 0.70 (*p*-value = 0.001). ED-based syndromic surveillances of fever were feasible for the early detection of hospital visits due to influenza.

## 1. Introduction

Influenza virus infections have substantially contributed to acute infectious respiratory diseases, causing illnesses from mild symptoms needing no medical attention to hospitalization or death [1]. Severe illnesses caused by influenza infections are more commonly observed in older people, infants, pregnant women, or people with underlying diseases. In countries with temperate climates, influenza epidemics occur during the winter, while in tropical regions, year-round infection patterns have been observed [2]. The disease burden of influenza infections was estimated to encompass 54.5 million lower respiratory infections, with 8.2 million severe cases and 145,000 deaths in 2017 by the Global Burden of Disease Study [3]. Another modeling study estimated the annual influenza-related mortality as 4.0–8.8 per 100,000 individuals, causing 409,111 influenza-infection-related respiratory deaths per year [4]. 

The global influenza surveillance and response system (GISRS) has been conducted through the World Health Organization to prevent and control influenza epidemics worldwide. A total of 127 countries have participated from 1952 to 2022 [5]. FluNet is an open internet-based system that consistently records population-level influenza in over 170 countries based on the GISRS and publicly summarizes global influenza activities [6]. As a public platform, FluNet supports monitoring, planning, and alerting the world to novel influenza epidemiology for seasonal, pandemic, and zoonotic influenzas [7]. In Korea, the Korea Disease Control and Prevention Agency (KCDA) operates an influenza surveillance system based on 200 influenza sentinel sites in primary clinics nationwide and a respiratory virus surveillance system which experiments with respiratory samples from 52 sentinel sites to identify nationwide influenza epidemics [8]. 

However, surveillance based on influenza-like illnesses takes time to report. To diagnose influenza, it is necessary to have a laboratory diagnosis of the antigen, RNA detection, or virus isolation through rapid tests, immunofluorescence assays, cell cultures, or RT-PCRs [9]. Time lags between suspected influenza cases and diagnoses are inevitable. Thus, one or two weeks of a delay in the identification of an influenza outbreak typically occur due to delayed sentinel site reports to the KCDA and laboratory tests [10], leading to delayed responses or preparation for an influenza epidemic. As an alternative, the syndromic surveillance of influenza and influenza-like-illnesses based on emergency department (ED) data, in which patients with acute diseases visit 24 h a day, 7 days a week, with real-time data collection, has been suggested to estimate influenza outbreaks. 

In Korea, clinical information during ED care is collected into a nationwide electronic database operated by the National Emergency Department Information System (NEDIS). Therefore, this study evaluated the use of chief complaint data from ED to detect the increment of influenza cases identified from nationwide medical service usage and developed a forecast model to predict the number of influenza patients accounted the daily number of ED visits due to fever [9]. 

## 2. Materials and Methods

### 2.1. Study Settings and Study Population

Secondary data analysis using the database of the National Health Insurance Service (NHIS) and the NEDIS was carried out. The NEDIS database includes clinical and administrative information for all patients visiting 167 institutions of regional and regional emergency medical centers nationwide [11]. The data from each emergency medical center were transferred to the NEDIS database. Despite the definition of “fever and respiratory” having been applied in other ED-based surveillance activities for influenza [12,13], we used fever as the only surveillance criteria for influenza. Although symptoms of influenza are varied and include fever, cough, sore throat, runny or stuffy nose, malaise, headache, or fatigue, most of the symptoms were non-specific and the proportion of ED visits with the above symptoms other than fever were not common. However, fever is an objective symptom and one of the common symptoms reported when visiting an ED [14,15]; thus, we applied fever as a surveillance criteria for influenza. In addition, previous studies applied fever as a surveillance criteria for influenza and showed good estimations. 

The definitions of fever included measured body temperature ≥ 38.0 °C at the ED and reported chief complaint by patients as fever [16]. We calculated the daily number of ED visits with fever between 1 January 2015 and 31 December 2019, based on the date patients visited the ED from NEDIS. The number of visitors to the ED on weekends increased (Appendix A and Appendix B). To minimize this effect, the number of daily ED visitors due to fever from 2015 to 2019 was defined as the moving average number of visitors for the last seven days up to that day. For example, the number of daily ED visits with fever on 7 January 2018 was defined as the average number from 1–7 January, and on 8 January 2018 it was defined as the average number from 2–8 January.

Because influenza surveillance in Korea collects influenza-like-illnesses only from 200 sentinel sites, we applied the National Health Insurance Service (NHIS) data to identify the number of influenza cases. Korea implements the NHIS, a mandatory national health insurance system covering 98% of the population. All medical services used with diagnostic codes and socio-demographic information of individuals are available in the NHIS data [17]. The NHIS data with qualifications and treatment history for each influenza patient from 1 January 2015 to 31 December 2019 were applied. Using health insurance claims data, patients who visited the hospital with the International Classification of Diseases-10 (ICD-10) codes of J09, J10, or J11 were defined as influenza cases [1]. Since this is claims-based data and separate claims could be generated for a single disease episode because different medical services could be consumed for single disease episode. When claim data were adjusted to episodes after applying a window of 100 days, although there were some differences between years, approximately 99% of all influenza patients experienced one outbreak [18]. Thus we applied the 100-day episode which grouped all claims within 100 days from first hospital visit due to influenza as single influenza cases as Byeon et al. did [18]. Based on this definition, the daily number of people visiting clinics due to influenza was counted.

### 2.2. Statistical Analyses

We developed a forecast model of daily influenza patients from the NHIS data based on the 7-day moving average of ED visits due to fever over the past days from the NEDIS data. Considering the characteristics of seasonal fluctuation of influenza, the seasonal autoregressive integrated moving average (ARIMA) was applied for forecasting time series in various medical sciences for short-term prediction [8,9,10]. NEDIS and NHIS data from 2015 to 2017 were used to develop a model. The parameters for seasonal ARIMA models include the order of autoregressive, degree of difference, and the order of moving average as non-seasonal components, and the order of seasonal autoregressive, seasonal difference, seasonal moving average, and the number of seasonal periods as seasonal components. The final model was selected based on the Akaike information criterion (AIC) and Bayesian information criterion (BIC). The model with the lowest AIC and BIC was selected. The NEDIS and NHIS data from 2018 to 2019 were applied to validate the model. In addition, the correlation coefficient was calculated between the daily 7-day moving average of ED visits with fever and the daily number of patients with influenza in the NHIS from 2015 to 2019. We conducted forecast accuracy of the model by the mean absolute percentage error (MAPE) which was defined as follows:$$\mathrm{MAPE}=\frac{\sum_{i=1}^{n\;}\frac{\left|Y_{i-}{\hat{Y}}_{i}\right|}{Y_{i}}\cdot 100}{n}$$
where $Y_{i}:\;\mathrm{oberved}\;\mathrm{ED}\;\mathrm{visits};\;{\hat{Y}}_{i}:\;\mathrm{predicted}\;\mathrm{ED}\;\mathrm{visits}.\;$ Analyses were performed with SAS (version 9.4; SAS Institute, Cary, NC, USA). 

## 3. Results

From 1 January 2015 to 31 December 2019, there were 29,142,229 ED visits according to the NEDIS data (Table 1). Of these, 8.4% had fever as a chief complaint, 25.0% had an initial body temperature at the ED of ≥38.0 °C, and 6.5% had both fever as a chief complaint and an initial body temperature at the ED of ≥38.0 °C.

It has been well established that the number of patients visiting the ED is higher during the weekend, and the proportion of less urgent or outpatient emergency visits is also higher during the weekend [19]. In addition to previous results, our dataset also identified an increased number of ED visits during the weekend (see Appendix A). We considered that by using a seven-day moving average, the number of ED visits would be minimally affected by the number of ED visits during the weekend with a real-time approach. The daily seven-day moving average of ED visits with fever from the NEDIS and the number of patients with influenza from the NHIS were plotted together in Figure 1.

The two data sources showed similar increasing and decreasing patterns and peaks. When the correlation of these two data was measured, the daily seven-day moving average of ED visits with fever and patients with influenza showed a moderate correlation with a Pearson’s correlation coefficient of 0.73 (*p*-value < 0.001, Figure 2). 

The data of 981,473 ED visits in 2015, 2016, and 2017 in the NEDIS were used to develop a forecast model. We measured the AIC and BIC values of three different forecasting ARIMA models. The AIC and BIC values of each model were as follows: AIC: 11245.0 and BIC: 11260.0 for the ARIMA(1,1,1)(0,0,1)₇ model, AIC: 11272.9 and BIC: 11287.8 for the ARIMA(1,1,1)(1,0,0)₇ model, and AIC: 11186.5 and BIC: 11206.5 for the ARIMA(1,1,1)(1,0,0) ₇ model. Although the AIC and BIC values of the ARIMA(1,1,1)(1,0,0) ₇ model are the lowest, the ACF and PACF of the residuals are outside the confidence limits. 

We selected the seasonal ARIMA (1,1,1) (0,0,1)₇ model to predict daily influenza patients based on the seven-day moving average of ED visits due to fever (AIC:11245.0 and BIC:11260.0). The parameters of the selected model are presented in Table 2. 

The equation is as follows: Visit_t_ = 1.67686y_t−2_ − 067686y_t−1_ − 0.21127↋_t−1_ + 0.45215↋_t−7_ − 0.095526↋_t−8_ + ↋_t_

**Table 2 ijerph-19-12954-t002:** Parameter estimation of the seasonal ARIMA(1,1,1)(0,0,1)₇ model to forecast hospital visits due to influenza based on a seven-day average ED visits with fever.

Parameter	Estimate	Standard Error	t Value	Approximate Pr > |t|	Lag
MA1,1	0.45215	0.02936	15.4	<0.0001	7
MA2,1	−0.21127	0.03849	−5.49	<0.0001	1
AR1,1	0.67686	0.02949	22.95	<0.0001	1

To validate the forecast model, we applied the model to the NEDIS and NHIS data from 2018 to 2019, with 898,835 ED visits due to fever and 5,778,838 patients with influenza. To check the forecast accuracy, we measured the MAPE value with the actual ED visits and the forecasted ED visits. The MAPE value shows a high prediction performance with the range of 2.2813–8.5615%(Table 3).

The daily seven-day and fourteen-day moving average of ED predicted visits with fever and the actual daily number of ED were plotted together in Figure 3. The Figure 3 shows similar trend and a high correlation with a Pearson’s correlation coefficient of 0.97 and 0.88.

Table 4 shows the correlation of the expected daily number of ED visits based on the seven-day moving average number of ED visits due to fever and the actual daily number of patients with influenza. The correlation coefficient was highest when the seven-day interval between the date of the ED visit and the number of patients had a correlation coefficient of 0.782 (*p*-value < 0.001). Up to the 14-day forecast interval, the correlation coefficient was higher than 0.7, with the exceptions of the eighth, ninth, and twelfth days’ forecast intervals; however, the correlation dropped to 0.62 when a 15-day interval was applied. Therefore, it is possible to predict the number of daily patients with influenza using the seven-day moving average of ED visits due to fever for up to two weeks.

## 4. Discussion

This study developed a forecast model of daily patients with influenza using real-time ED-based syndromic surveillance by monitoring the number of patients visiting the ED due to fever. When the model was validated using the 2018–2019 data, the correlation between the expected number of daily hospital visits due to influenza and the actual number was moderate to high until the 14-day interval. It suggests that monitoring the number of ED visits due to fever could predict the daily number of patients with influenza 14 days in advance; thus, it could be useful to predict the surge of patients with influenza and decide whether to provide a warning to the public.

Syndromic surveillance has effectively provided information during seasonal and sporadic events to confirm the absence of any public impact for both infectious and non-infectious diseases. Syndromic surveillance could provide information more quickly than laboratory systems or confirmation tests and augment the laboratory surveillance systems for infectious diseases [3]. There have been examples of early detection of influenza compared to traditional disease-specific systems in the United States. Influenza has been suggested to be one of the diseases in which syndromic surveillance would be the most useful [20].

The data for syndromic surveillance are mainly collected through patient contact with a primary physician. However, recently various non-healthcare-related data sources, such as social media or internet search data, have been used for syndromic surveillance [21,22]. Of the various sources of syndromic surveillance, ED data have become an important source because acute cases are collected in ED. Several countries have already developed an electronic system to collect ED records, and it provides and helps complete immediate data collection, which can rapidly be made available for public health surveillance [23]. During the coronavirus disease (COVID-19) pandemic, syndromic surveillance was useful in detecting anomalous clusters and hospitalization numbers [24,25]. Syndromic surveillance based on respiratory ED visits was also valuable in monitoring COVID-19 in other settings [26].

Syndromic surveillance systems for influenza and influenza-like-illness (ILI) from ED data are becoming more prevalent worldwide and more accepted as a measurement of influenza outbreaks yearly [27]. Patients’ chief complaints and diagnosis codes are the most often used data for ED-based surveillance. Studies suggested that diagnostic codes showed better sensitivity than chief complaints, especially for lower respiratory diseases [27,28]. In addition, a recent review study suggested that the format of information of the syndromic indicators of EDs in Korea for influenza was the diagnosis code [23,29]. However, in the ED, time to obtain discharge diagnosis codes takes time, from the day of the ED visit to admission to the hospital or discharge. In contrast, the chief complaint is reported upon arrival in EDs and body temperature is measured for all ED patients during the initial triage. Then, the information is transferred to the central NEDIS system directly without delay. One study identified the usefulness of the daily number of ED visits with fever based on measured body temperature upon arrival in EDs to forecast infectious respiratory disease outbreaks using the NEDIS data [29]. As a result of this study, Kim et al. showed that the syndromic surveillance based on the daily number of ED visits with fever for infectious respiratory disease outbreaks would be feasible [29,30]. The two distinctions between this study and the study by Kim et al. were as follows: (1) This study included nationwide ED visits due to fever from the NEDIS data and patients with influenza data from the NHIS. These were two independent datasets. The study by Kim et al. applied ED visits due to fever and the final diagnosis of respiratory infectious diseases from the NEDIS data. (2) Kim et al. set the threshold as the upper 95% for the expected number of the forecast model for alarm; however, this study showed a correlation between the expected number from the seasonal ARIMA forecast model and the actual number of patients with influenza from the NHIS data. Therefore, we believe that this model could be useful in deciding whether to provide a warning to the public and it can be used to predict the number of patients with influenza in the next 14 days. Thus, the expected number of patients with influenza from the forecast model of this study would help public health sectors and hospitals prepare for the control and care of influenza during the winter season.

In Korea, the usefulness of the NEDIS data for syndromic surveillance was evaluated for epidemic Keratoconjunctivitis and meningitis [30], acute diarrheal syndromes due to mass food poisoning [31], and acute toxic exposures [32]. This study could add evidence of the usefulness of the ED-based syndromic surveillance for influenza using the NEDIS data.

Several limitations of this study should be addressed. First, the NHIS data is claim-based, and the identified patients with influenza were not confirmed cases based on laboratory diagnoses. Thus, the number of patients with influenza from the NHIS data would only be suspected cases. Second, the input of the chief complaint to the NEDIS database is mandatory for emergency medical centers but optional for emergency medical agencies. Thus, the chief complaint may have been missed in several patients. However, the input of vital signs, including body temperature, is mandatory for emergency medical centers and emergency medical agencies. Despite missing values in chief complaints, this study used both chief complaints and measured body temperature to identify cases with fever to reflect real ED data to forecast influenza patients. Third, the quality of the NEDIS data would be different across the institutions. However, quality assessment and certification of the data quality of the NEDIS database are performed regularly by the National Emergency Medical Center. Fourth, this study developed a forecast model of influenza before the COVID-19 pandemic. With the updated data, the validity of this model during the COVID-19 pandemic needs to be assessed. Further, a forecast model in combination with an epidemic of the new emerging infectious diseases needs to be developed. Fifth, although influenza outbreak alerts are based on sentinel surveillance, sentinel surveillance covers 200 sentinel sites; thus, we applied NHIS data, which covers the whole population in Korea. However, the trend of the number of patients with influenza would be different between the sentinel surveillance data and the NHIS data. Despite the limitations, in Korea, syndromic surveillance using ED data developed with national coverage [23]; thus, selection bias would be minimal. In addition, the NHIS covers almost all of the population of Korea [17]; therefore, the trend of patients with influenza would be monitored with nationwide coverage.

## 5. Conclusions

This study developed a forecast model for the number of patients with influenza visiting hospitals using an ED-based syndromic surveillance of fever in Korea. The expected number of patients with influenza by the seasonal ARIMA model using ED data was highly correlated with the observed number of patients with influenza in the NHIS data, and the forecasted number showed a good correlation with the actual number up to two weeks in advance. Therefore, ED-based syndromic surveillance using the NEDIS database could identify influenza surges earlier than the current system, helping us better prepare for an epidemic.

## Figures and Tables

**Figure 1 ijerph-19-12954-f001:**
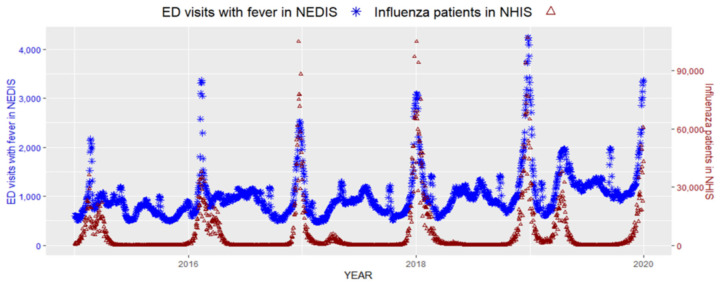
Seven-day moving average of ED visits with fever and daily patients of influenza in the NHIS.

**Figure 2 ijerph-19-12954-f002:**
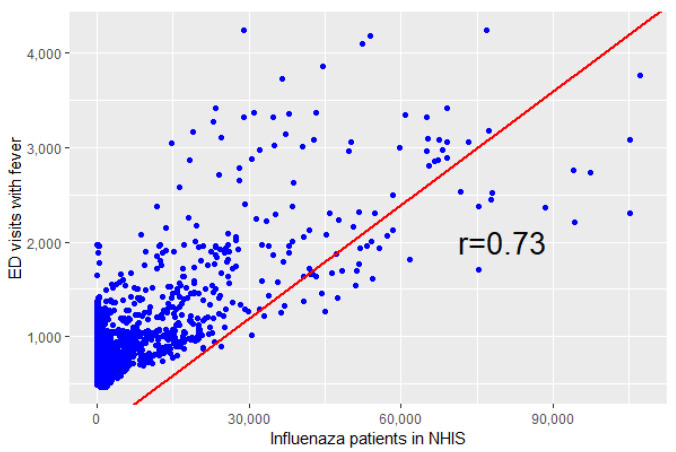
Correlation coefficient between the seven-day moving average of ED visits with fever and daily patients of influenza in the NHIS.

**Figure 3 ijerph-19-12954-f003:**
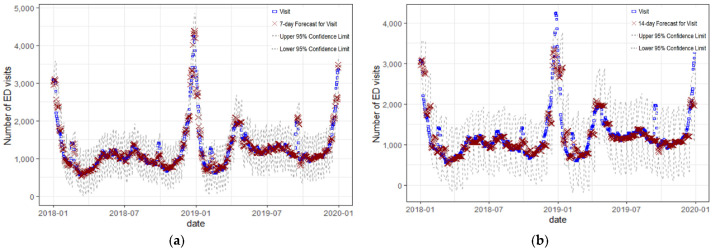
(**a**) The seven-day moving average of ED visits with fever and the prediction of the seven-day average ED visits with fever (**b**) The seven-day moving average of ED visits with fever and the prediction of 14-day moving average ED visits with fever.

**Table 1 ijerph-19-12954-t001:** Number of ED visits, ED visits with fever, and ED visits with final diagnosis of influenza.

Year	Total	2015	2016	2017	2018	2019
Total ED visits	29,142,229	5,359,831	5,823,780	5,813,188	5,998,742	6,146,688
ED visits with chief complaint of fever	2,447,446 (8.4%)	398,237 (7.4%)	505,334 (8.7%)	419,465 (7.2%)	544,454 (9.1%)	579,956 (9.4%)
Body temperature ≥ 38.0°C at the ED	7,298,957 (25.0%)	1,346,799 (25.1%)	1,609,449 (27.6%)	1,458,243 (25.1%)	1,545,582 (25.8%)	1,338,884 (21.8%)
Both fever as a chief complaint and body temperature ≥ 38.0 °C at the ED	1,880,308 (6.5%)	285,230 (5.3%)	374,006 (6.4%)	322,237 (5.5%)	429,432 (7.2%)	469,403 (7.6%)
Patients with influenza in the NHIS	11,182,104	1,141,514	2,772,409	1,489,343	3,505,807	2,273,031

**Table 3 ijerph-19-12954-t003:** The result of fitted ARIMA(1,1,1)(0,0,1)₇ model to forecast ED visits based on actual ED visits with fever.

Number of Day in Forecast	1-Day	2-Day	3-Day	4-Day	5-Day	6-Day	7-Day
MAPE(%)	2.2813	3.4674	4.0205	4.6546	7.1822	8.5615	6.4605

**Table 4 ijerph-19-12954-t004:** Correlation between the prediction of seven-day moving average ED visits with fever and daily number of influenza patients in NHIS from 2018 to 2019. We estimated the seasonal ARIMA(1,1,1)(0,0,1)₇ model to forecast hospital visits due to influenza based on a seven-day average ED visits with fever.

Number of Day in Forecast	Correlation Coefficient	*p*-Value
1-day	0.772	<0.0001
2-day	0.763	<0.0001
3-day	0.753	<0.0001
4-day	0.733	<0.0001
5-day	0.765	<0.0001
6-day	0.740	<0.0001
7-day	0.782	<0.0001
8-day	0.685	<0.0001
9-day	0.684	<0.0001
10-day	0.743	<0.0001
11-day	0.720	<0.0001
12-day	0.666	<0.0001
13-day	0.723	<0.0001
14-day	0.775	<0.0001
15-day	0.624	<0.0001
16-day	0.666	<0.0001
17-day	0.647	<0.0001
18-day	0.602	<0.0001
19-day	0.594	<0.0001
20-day	0.577	<0.0001

## Data Availability

The datasets used and analysed during the current study are available from the National Health Insurance Service (NHIS) data and the National Emergency Department Information System (NEDIS) data.

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
