# Peer review of "Forecasting Hospital Visits Due to Influenza Based on Emergency Department Visits for Fever: A Feasibility Study on Emergency Department-Based Syndromic Surveillance"

_ijerph, 2022, doi:10.3390/ijerph191912954_

Round 1

Reviewer 1 Report

Need to improve the readability with relevant context to be deployed continually. Whilst the topic is interesting with valid datasets, it requires to present key findings and achievements in an efficient way for presentation of the idea.

Author Response

Point 1: Need to improve the readability with relevant context to be deployed continually. Whilst the topic is interesting with valid datasets, it requires to present key findings and achievements in an efficient way for presentation of the idea.

Response 1:

Thank you for this comment. After revising the manuscript according to the comments from reviewers 1-4, we further engaged an English editing service to increase the readability of the manuscript. The certificate of the English editing service is attached below.

Reviewer 2 Report

Dear Authors,

It is an honor to be able to review this manuscript.

Hopefully some comments could improve the quality of the paper and wishing you all the success.

Regards,

Viewer

Author Response

Point 1: please add reference indicating death caused by influenza. 

Response 1:

Thank you for your suggestion. We have added the reference as follows:

“Influenza virus infection has substantially contributed to acute infectious respiratory diseases, causing illnesses from mild symptoms needing no medical attention to hospitalization or death [1].”(line 40)

Point 2: Before the discussion section, I think it will be better to add accuracy or calculate the error either MAD, MSE, or MAPE to more justify the result and convince the reader that the prediction is reliable.

Response 2:

Thank you for your comments.

Accordingly, we have added the results of the MAPE of the final ARIMA(1,1,1)(0,0,1)₇ model as an updated Table 3, and have described it as follows:

“To validate the forecast model, we applied the model to the NEDIS and NHIS data from 2018 to 2019, with 898,835 ED visits due to fever and 5,778,838 patients with influenza. To check the forecast accuracy, we measured the MAPE value with the actual ED visits and forecasted ED visits. The MAPE value showed a high prediction performance with a range of 2.2813%–8.5615% (Table 3).“ (line185-190)

Table 3. Result of the fitted ARIMA(1,1,1)(0,0,1)₇ model for forecasting hospital visits due to influenza based on a 7-day average number of ED visits of patients with fever.

Number of day in forecast

1-day

2-day

3-day

4-day

5-day

6-day

7-day

MAPE

2.2813%

3.4674%

4.0205%

4.6546%

7.1822%

8.5615%

6.4605%

Reviewer 3 Report

Summary

Thank you for your efforts for generating important research work for forecasting potential number of patients with influenza that leverages the real time monitoring of the emergency department visits for patients with fever. This kind of surveillance process can help in managing the public health by mitigating the time lag associated with the diagnosis. The authors of the manuscript have utilized the National Emergency Department Information System (NEDIS) database as input and the National Health Insurance Service (NHIS) database as the ground truth information for developing a model intended to forecast influenza cases. The model was developed using the data collected between 2015-2017 and were validated on data collected between 2015 to 2019 for the NEDIS and NHIS databases.  

Overall, the developed model can add to the knowledgebase on utility of model leveraging patients with fever visiting emergency department for influenza. This information combined with additional analysis for other disease areas with fever as a sign can help develop more robust methods for disease monitoring for public health.

Please find the comments associated with the manuscript in the following section.

Comments:

Abstract

Line 31: The correlation coefficient for 3-day forecast was reported to be 0.7534 and line 147 states that the correlation coefficients were higher than 0.75. The information in the abstract should be corrected for it.

Introduction

Line 46: Please provide a reference for this study.

Line 50: Please update the reference for the page that contains this information.

Line 54: Please update the reference numbering. Also, please check if the sentence is specifically for the FluNet platform.  

Line 56-58: Please update the font for consistency.

Line 64: With regard to the reference provided, is it for the inferential purposes or is it providing the information related to KCDA?

Materials and Methods

Line 81-82: It would be good to provide insight/reasoning for the use of fever as the only surveillance criteria for influenza.

Line 93-94: It seems like there is a limitation because the data is collected only from the 200 sentinel sites and to overcome that NHIS data was used. It would be good to provide clarification on the limitation in this sentence.

Line 98. Please clarify the following sentence: “The NHIS data from January 1, 2015, to December 31, 2019, were applied.”

Line 104-105: Based on the previous line, all the claims within 100 days of window period are considered as one. It would be helpful for readers if you can provide more information on how the daily number of cases were determined considering this information.

Results

Line 131-132: Please add the rational for using 7-day moving day average as opposed to any other duration. Was it due to an analysis performed to select it or was it based on the existing literature sources?

Line 137-136: It would be informative to users to provide the additional details of how the specific model parameters were selected, any information on the additional models that might have been evaluated, reasoning for why the specific model was selected in comparison the other evaluated models.

Line 138: It would be good to use a consistent terminology for SBC ( BIC is used in line 117)

Figure 1: Updating the existing axis labels with more informative ones would ease the information communication to the readers.

Line 156: It would be helpful to also provide the forecasting equation for the selected model.

Figure 3. It would be helpful to add confidence limits to the forecasts.  

Table 3: It would be good to provide the information for all the forecasts up to the 14-day forecast and may be even two or three days beyond to show how the performance declines.

Additional information: It would be helpful to provide additional error metrics to gain insights into how forecasts differ from the daily patients of influenza in NHIS.

Discussion

Line 224- 227:  The model can potentially be used for the warning purposes, but it should go through appropriate analysis of the predictive performance and warning generation process before this claim can be made.  I would suggest to either modify the language or provide additional information to support the claim, whichever is applicable. 

Appendix

Figures: Please provide descriptions for the figures presented and more informative y-axis labels.  

Author Response

Response to Reviewer 3 Comments

Point 1: (Abstract Line 31) The correlation coefficient for 3-day forecast was reported to be 0.7534 and line 147 states that the correlation coefficients were higher than 0.75. The information in the abstract should be corrected for it.

Response 1:

We apologize for the typographical errors in the Abstract section. We have corrected “The correlation coefficient between the number of ED visits and the number of patients with influenza in the NHIS up to 14 days before the forecast was around 0.75 or higher (P-value = 0.001)” to “The correlation coefficient between the number of ED visits and the number of patients with influenza in the NHIS up to 14 days before the forecast was approximately 0.70 (P-value = 0.001)”.(line 31-33)

Point 2: (Introduction Line 46) Please provide a reference for this study.

Response 2:

Thank you for your comment. We have added a reference for this study:

“The disease burden of influenza infection was estimated to be 54.5 million lower respiratory infections, with 8.2 million severe cases and 145,000 deaths in 2017, according to the Global Burden of Disease Study[2].”(line 43-46)

Point 3: (Introduction Line 50) Please update the reference for the page that contains this information.

Response 3:

We apologize for the typographical errors. We have corrected “A total of 115 countries have participated from 1952 to 2015” to “A total of 127 countries have participated from 1952 to 2022”. (Line 51)

Additionally, the reference was corrected to: "Global influenza surveillance and response system." 2022. https://www.who.int/news/item/03-02-2022-2022-celebrating-70-years-of-gisrs-(the-global-influenza-surveillance-and-response-system). (accessed on 18 Sep 2022).

You can find the corrected version directly in the reference list.

Point 4: (Introduction Line 54) Please update the reference numbering. Also, please check if the sentence is specifically for the FluNet platform.

Response 4:

We apologize for the typographical errors. We have corrected the reference numbering.

In addition, to avoid confusion, we have corrected the sentence as follows:

“FluNet is an open internet-based system that consistently records population-level influenza in over 170 countries based on the GISRS and publicly summarizes global influenza activities [6]. As a public platform, FluNet supports the monitoring, planning, and alerting the world of novel influenza epidemiology for seasonal, pandemic, and zoonotic influenza [7].”(line 51-55)

Point 5: (Introduction Line 56-58) Please update the font for consistency.

Response 5:

We apologize for the typographical errors. We have corrected the font for consistency.

Point 6: (Introduction Line 64) With regard to the reference provided, is it for the inferential purposes or is it providing the information related to KCDA?

Response 6:

Although the KCDA has not officially reported a delay, a possible delay of not only the KCDA reporting system but also other reporting systems has been continuously suggested.

You can find it at the link below:

List | Korea Influenza Weekly Report | Archives : KDCA

In addition, influenza has been reported in the sentinel surveillance system in Korea. According to the Infectious Disease Control and Prevention Act in Korea, sentinel surveillance systems request sentinel clinics to report the disease once a week, thus a 1- or 2-week delay could be expected.

Point 7: (Materials and Methods Line 81-82) It wo uld be good to provide insight/reasoning for the use of fever as the only surveillance criteria for influenza.

Response 7:

Thank you for your comment.

We have added the following sentences:[1]

“Although the symptoms of influenza vary and include fever, cough, sore throat, runny or stuffy nose, malaise, headache, or fatigue, most of the symptoms are non-specific, and the proportion of ED visits due to symptoms other than fever was not common. However, fever is an objective symptom and one of the most common symptoms that leads to ED visits [13, 14]; thus, we used fever as a surveillance criterion for influenza. In addition, previous studies that applied fever as a surveillance criterion for influenza have shown good estimations.”(line 86-90)

Point 8: (Materials and Methods Line 93-94) It seems like there is a limitation because the data is collected only from the 200 sentinel sites and to overcome that NHIS data was used. It would be good to provide clarification on the limitation in this sentence.

Response 8:

We added the following point in the limitation section of the manuscript:

“Fifth, although influenza outbreak alerts are notified based on the sentinal surveillance, the coverage of the sentinal surveillance is limited to 200 sentinel sites; thus we applied NHIS data, which covers the entire population in Korea. However, the trend in the number of patients with influenza would differ between the sequential surveillance data and the NHIS data.“ (line 293-297)

Point 9: (Materials and Methods Line 98) Please clarify the following sentence: “The NHIS data from January 1, 2015, to December 31, 2019, were applied.”

Response 9: Thank you for your comment.

We have made the following correction:

“The NHIS data with qualifications and treatment history of each influenza patient from January 1, 2015, to December 31, 2019, were applied.”(line 107-109)

Point 10: (Materials and Methods Line 104-105) Based on the previous line, all the claims within 100 days of window period are considered as one. It would be helpful for readers if you can provide more information on how the daily number of cases were determined considering this information.

Response 10:

Thank you for your comment.

We have added the following content to the Methods section to help readers understand our approach:

“Since the data included is claims-based, separate claims could be generated for a single disease episode as different medical services could be utilized for single disease episodes. When claims data were adjusted to episodes after applying a window of 100 days, although there were some differences between the years, approximately 99% of all influenza patients experienced one outbreak [17]. Thus, we applied a 100-day episode that grouped all claims within 100 days from the first hospital visit due to influenza as a single influenza case, similar to Byeon et al.”(line108-115)

Point 11: (Results Line 131-132) Please add the rational for using 7-day moving day average as opposed to any other duration. Was it due to an analysis performed to select it or was it based on the existing literature sources?

Response 11:

Thank you for your comments. It has been well established that the number of patients visiting the ED is higher during the weekend and the proportion of less urgent or outpatient emergency visits is also higher during the weekend[18]. In addition to previous results, our dataset also identified an increased number of ED visits during the weekend (see Appendix A). We considered that by using a 7-day moving day average, the number of ED visits would be minimally affected by the number of ED visits during the weekend with a real-time approach. In addition, we have added the following explanation in the Results section:

“It has been well established that the number of patients visiting the ED is higher during the weekend, and the proportion of less urgent or outpatient emergency visits is also higher during the weekend[18]. In addition to previous results, our dataset also identified an increased number of ED visits during the weekend (see Appendix A). We considered that by using a 7-day moving day average, the number of ED visits would be minimally affected by the number of ED visits during the weekend with a real-time approach.”(line 150-159)

Point 12: (Results Line 137-140) It would be informative to users to provide the additional details of how the specific model parameters were selected, any information on the additional models that might have been evaluated, reasoning for why the specific model was selected in comparison the other evaluated models.

Response 12:

Thank you for your suggestion.

We have added the following sentences:

“We measured the AIC and SBC values of three different forecasting ARIMA models. The AIC and SBC values of each model were as follows: AIC: 11245.0 and SBC: 11260.0 for the ARIMA(1,1,1)(0,0,1) ₇ model, AIC: 11272.9 and SBC: 11287.8 for the ARIMA(1,1,1)(1,0,0) ₇ model, and AIC: 11186.5 and SBC: 11206.5 for the ARIMA(1,1,1)(1,0,0) ₇ model.
Although the AIC and SBC values of the ARIMA(1,1,1)(1,0,0) ₇ model are the lowest, the ACF and PACF of the residuals are outside the confidence limits.”(line 168-173)

Point 13: (Results Line 138) It would be good to use a consistent terminology for SBC ( BIC is used in line 117)

Response 13:

Thank you for your comment.

We have corrected “SBC” to “BIC”.

Point 14: (Results Figure 1): Updating the existing axis labels with more informative ones would ease the information communication to the readers.

Response 14:

Thank you for your comment.

We have corrected the axis labels.

Point 15: (Results Line 156) It would be helpful to also provide the forecasting equation for the selected model.

Response 15:

Thank you for your comment. We have added the following sentence(line178-180):

“The equation became

Visitt = 1.67686yt-2 - 067686yt-1 - 0.21127↋t-1 + 0.45215↋t-7 - 0.095526↋t-8 + ↋t

Point 16: (Results Figure 3) It would be helpful to add confidence limits to the forecasts.

Response 16:

Thank you for your comment.

We have added confidence limits to the forecasts.

Point 17: (Results Table 3) It would be good to provide the information for all the forecasts up to the 14-day forecast and may be even two or three days beyond to show how the performance declines.

Response 17:

Thank you for your comment.

We performed predictions up to day 20 and checked the changes in the correlation. Except for the 8th, 9th, and 12th days, it was confirmed that the correlation coefficient was maintained at 0.7 until the 14th prediction.

Table 4 shows day 1 to day 20 forecasts in 1-day intervals. Additionally, we have corrected the following sentence:

“Up to the 14-day forecast interval, the correlation coefficient was higher than 0.7 with the exceptions of the 8-day, 9-day, and 12-day forecast intervals; however, the correlation dropped to 0.62 when a 15-day interval was applied. Therefore, it is possible to predict the number of daily patients with influenza using the 7-day moving average of ED visits due to fever for up to 2 weeks.”(line 206-208)

Point 18: (Results Additional information) It would be helpful to provide additional error metrics to gain insights into how forecasts differ from the daily patients of influenza in NHIS.

Response 18: Thank you for this comment. Accordingly, we have added the results of the MAPE as one of the error metrics for the final ARIMA(1,1,1)(0,0,1)₇ model in Table 3 and described this as follows:

“To validate the forecast model, we applied the model to the NEDIS and NHIS data from 2018 to 2019, with 898,835 ED visits due to fever and 5,778,838 patients with influenza. To check the forecast accuracy, we measured the MAPE value with the actual and forecasted ED visits. The MAPE value showed a high prediction performance with a range of 2.2813%–8.5615% (Table 3). “

Point 19: (Discussion Line 224- 227) The model can potentially be used for the warning purposes, but it should go through appropriate analysis of the predictive performance and warning generation process before this claim can be made. I would suggest to either modify the language or provide additional information to support the claim, whichever is applicable.

Response 19:

Thank you for your comment. We have revised the sentence as follows:

“It suggests that monitoring the number of ED visits due to fever could predict the daily number of patients with influenza 14 days in advance; thus, it could be useful to predict the surge of patients with influenza and decide whether to provide a warning to the public.”(line 224-226)

“Therefore, we believe that this model could be useful in deciding whether to provide a warning to the public and it can be used to predict the number of patients with influenza in the next 14 days.”(line268-269)

Point 20: (Appendix Figures) Please provide descriptions for the figures presented and more informative y-axis labels.

Response 20:

Thank you for your comment. We have corrected the Appendix Figures.

Reviewer 4 Report

A feasibility study on emergency department based syndromic surveillance to forecast hospital visits due to influenza is presented in this manuscript. The manuscript is interesting and well written. The following are my suggestions to further improve the manuscript. 

References for the literature in the year 2022 are missing in the manuscript. Authors have expected to add recent references. 

Peason coefficient should be plotted in the figure 2 otherwise the figure is not self explanatory. 

Figure qualities are poor. Should be improved. Cannot fathom the information.

Predictive performance of the model can be presented in a table comparing essential parameters. 

Author Response

Point 1: References for the literature in the year 2022 are missing in the manuscript. Authors have expected to add recent references.

Response 1:

Thank you for your suggestion.

We have added references in the manuscript from the year 2022.(line 41)

Point 2: Pearson coefficient should be plotted in the figure 2 otherwise the figure is not self explanatory.

Response 2:

Thank you for your comment. We have corrected Figure 2.(line 164)

Round 2

Reviewer 1 Report

response satisfies the feedback

Author Response

Point 1: response satisfies the feedback.

Response 1:

 We are grateful to you for your time and constructive comments on our manuscript.

Reviewer 3 Report

Thank you for addressing the previous comments. The manuscript overall looks great! Please address few minor aspects in the instances listed below.

Line 33: It is still unclear how the value of 0.70 was derived which is used in the abstract as no other reference in the rest of the manuscript can be found for this value.

Line 92: Please provide the reference for the previous studies.

Line 192-197: There is a typo in the model listed here.

Author Response

Point 1: (Line 33) It is still unclear how the value of 0.70 was derived which is used in the abstract as no other reference in the rest of the manuscript can be found for this value.

Response 1:

We apologize for the unclear of the Abstract section.

We have corrected “The correlation coefficient between the number of ED visits and the number of patients with influenza in the NHIS up to 14 days before the forecast was around 0.70 (P-value = 0.001)” to “The correlation coefficient between the number of ED visits and the number of patients with influenza in the NHIS up to 14 days before with the exceptions of the 8-day, 9-day, and 12-day the forecast was higher than 0.70 (P-value = 0.001).” (Line 33-35)

You can find this value in the Table4.

Point 2: (Line 92) Please provide the reference for the previous studies.

Response 2:

Thank you for your suggestion. We have added the reference as follows:

The definitions of fever included measured body temperature ≥ 38.0°C at the ED or reported chief complaint by patients as fever[16].(Line92-93)

16."Definitions of symptoms for reportable illnesses." 2022.

https://www.cdc.gov/quarantine/air/reporting-deaths-illness/definitions-symptoms-reportable-illnesses.html. (accessed on 3 Oct 2022).

Point 3: (Line 192-197) There is a typo in the model listed here.

Response 3:

Thank you for your comment.

We have made the following correction:

Table 3. the result of fitted ARIMA(1,1,1)(0,0,1)₇ model to forecast ED visits based on actual ED visits with fever.

Number of day in forecast

1-day

2-day

3-day

4-day

5-day

6-day

7-day

MAPE(%)

2.2813

3.4674

4.0205

4.6546

7.1822

8.5615

6.4605
